# The Role of the Left Inferior Frontal Gyrus in Introspection during Verbal Communication

**DOI:** 10.3390/brainsci13010111

**Published:** 2023-01-07

**Authors:** Ayumi Yoshioka, Hiroki C. Tanabe, Eri Nakagawa, Motofumi Sumiya, Takahiko Koike, Norihiro Sadato

**Affiliations:** 1Department of Cognitive and Psychological Sciences, Graduate School of Informatics, Nagoya University, Nagoya 464-8601, Japan; 2Japan Society for the Promotion of Science, Tokyo 102-0083, Japan; 3Division of Cerebral Integration, Department of System Neuroscience, National Institute for Physiological Sciences (NIPS), Okazaki 444-8585, Japan; 4Research Organization of Science and Technology, Ritsumeikan University, Kusatsu 525-8577, Japan

**Keywords:** conversation, hyperscanning functional magnetic resonance imaging, inferior frontal gyrus, mentalizing network, introspection, superior temporal gyrus

## Abstract

Conversation enables the sharing of our subjective experiences through verbalizing introspected thoughts and feelings. The mentalizing network represents introspection, and successful conversation is characterized by alignment through imitation mediated by the mirror neuron system (MNS). Therefore, we hypothesized that the interaction between the mentalizing network and MNS mediates the conversational exchange of introspection. To test this, we performed hyperscanning functional magnetic resonance imaging during structured real-time conversations between 19 pairs of healthy participants. The participants first evaluated their preference for and familiarity with a presented object and then disclosed it. The control was the object feature identification task. When contrasted with the control, the preference/familiarity evaluation phase activated the dorso-medial prefrontal cortex, anterior cingulate cortex, precuneus, left hippocampus, right cerebellum, and orbital portion of the left inferior frontal gyrus (IFG), which represents introspection. The left IFG was activated when the two participants’ statements of introspection were mismatched during the disclosure. Disclosing introspection enhanced the functional connectivity of the left IFG with the bilateral superior temporal gyrus and primary motor cortex, representing the auditory MNS. Thus, the mentalizing system and MNS are hierarchically linked in the left IFG during a conversation, allowing for the sharing of introspection of the self and others.

## 1. Introduction

Sharing one’s experiences with others is a fundamental human ability [1] that can be partially mediated by conversation (in this case, verbal communication) [2]. Conversations help us share our subjective experiences by verbalizing thoughts and feelings otherwise inaccessible to others. However, its neural underpinnings remain to be elucidated.

Conversation is a complicated process that requires joint action and is characterized by interactive alignment [3]. Alignment is defined as the tendency of conversational partners to choose the same words, syntactic structures, or phonology [3]. According to this account, a conversation is successful when participants come to an understanding of what they are talking about in the same way [4]. During the conversation, conversational partners construct mental models of the situation under discussion [5]; successful conversation occurs when these models are aligned and shared [4], helping impart one’s worldly experiences. Menenti et al. [4] argued that conversational partners tend to become aligned at different levels of linguistic representation and therefore find it easier to perform this through joint activity than through the individual activities of speaking or listening [3]. Moreover, Menenti et al. [4] suggested that alignment is a form of imitation.

Imitation is understood as a prediction/control relation or as a forward/inverse internal model [6,7,8]. The internal model defines the relations between motor commands and their sensory consequences [9]. Imitators decode the visual or auditory information presented by imitatees into their motor representation to generate a series of motor commands. However, decoding the actions of others is an ill-posed problem because the same sensory input can have several causes [10]. The predictive coding account suggests that both the optimization of self-action and inference of others’ actions require a forward model or top-down prediction [10]. That is, the same forward model used to predict the sensorial effects of one’s own actions can also be used as a constraint for decoding the actions of others [10,11]. The inverse model of the imitator is an inversion of the imitatee’s forward model with the imitator’s forward model as a constraint. As the imitator’s motor outputs reflect the imitator’s forward model, the iteratively updating processes result in shared forward internal models between the interacting agents [12,13]. Thus, a conversation is an alignment process leading to the shared forward internal model.

Predictive coding theory [14,15,16] proposes that the comparison of the top-down signal of the mental model with the lower representation generates a prediction error, which is fed back up the hierarchy to update higher representations. This recursive exchange of signals suppresses the prediction error at every level to provide a hierarchical explanation for sensory inputs [12]. The visual mismatch responses reflect the prediction errors obtained from a formal Bayesian model [17], supporting the theory. In language processing, predictive coding in the brain’s response to language is domain-specific, which is sensitive to the hierarchical structure [18]. The higher level of the hierarchical structure represents subjective experiences such as introspection (the capacity to attend to one’s thoughts and feelings) [19], which is represented by the mentalizing network [20]. Thus, alignment represents the shared processing of the forward internal models at multiple levels between conversational partners, driven by prediction errors at each level. Through this hierarchical structure, the mismatch between what you and your partner uttered drives the conversation, ultimately leading to understanding each other.

A pertinent example of transmitting introspection during conversation is sarcasm. Sarcasm mediates an implicit criticism of the speaker by provoking negative emotions with disapproval, contempt, and scorn [21]. During a conversation, sarcasm is perceived as a multi-layered incongruity between the context, content, and prosody of the utterance [22]. Uchiyama et al. [23] found that the mentalizing network and left inferior frontal gyrus (IFG) were activated during sarcasm comprehension and argued that the mentalizing network represents the speaker’s attitude toward the situation. They concluded that the left IFG integrates semantic and mentalizing processes during sarcasm detection. Matsui et al. [24] and Nakamura et al. [25] also showed that during sarcasm comprehension, the left IFG is related to detecting incongruity between the context, content, and prosody of the utterance. Thus, within the hierarchical predictive coding scheme, activity in the left IFG may be involved in incongruity detection during pragmatic tasks that require non-literal information. This suggests that during a conversation, a self–other comparison of non-literal information related to introspection may occur in the left IFG.

The IFG, along with the inferior parietal lobule (IPL), is a core component of the mirror neuron system (MNS) [26], and it may be well-suited to receive bottom-up signals from others. Recent models of speech processing [27,28] are based on the dual-stream model [29], which postulates that the ventral and dorsal streams converge in the IFG as follows: the dorsal stream targets the Brodmann Area (BA) 44, whereas the ventral stream ends in BA 45 ([30], in review). Thus, the left IFG “plays an important role in the transformation of highly processed auditory sensory information into motor-articulatory signals and vice versa” [30]. As such, the “mirror” function is critical in conversation.

The left IFG is a core part of the MNS, is involved in linguistic processes, and codes mismatches during linguistic tasks. Based on this, we hypothesized that the comparison of introspection during face-to-face conversation is mediated by the interaction between the mentalizing system and MNS through the left IFG.

In the present study, we modified the hyperscanning fMRI setting utilized in a previous study [2]. Participants were required to determine their preference for and familiarity with a presented target and to disclose them to a paired participant in another fMRI scanner. The attitude (preference/familiarity) toward the shared target was contrasted with the object feature identification task, in which the participant attended to the feature of the object during the evaluation phase and then uttered that feature to the partner, thereby requiring minimum introspection. To include the match/mismatch conditions, it was possible for the presented object to differ during the object feature identification task. The participants were notified of these conditions.

We expected that the self–other comparisons of the mental model of preference/familiarity during conversation would lead to a mismatch-related activity caused by mismatches between the disclosed preference and familiarity. Furthermore, we inferred that the bottom-up signal is mediated by the auditory MNS, which allows lower-level alignment. This is because the bottom-up information of the conversational partner is transferred across the predictive coding hierarchy from the lower level (for the alignment of the sensorimotor component of the utterance) to the higher level (for the alignment of the introspection; that is, the contents of the utterance). We performed psychophysiological interaction (PPI) analysis with the left IFG as the seed under the premise that the functional connectivity (FC) of the left IFG with the top-down and bottom-up signals would be enhanced during the disclosure of the introspection.

## 2. Methods

### 2.1. Participants

Thirty-eight healthy right-handed [31] adult volunteers (18 males, 20 females; 19 pairs, age 20.7 ± 1.91, mean ± standard deviation [SD] years) from the general population participated in this study. All participants were native Japanese-speaking people participating in ordinary social interactions such as schooling and housekeeping. Before the experiment, we paired same-sex participants who had never seen each other previously. None of the participants had a history of neurological or psychiatric illness. The ethics committee of the National Institute for Physiological Sciences (NIPS, Okazaki, Japan) approved the protocol. All participants gave written informed consent to participate in the study. The fMRI experiment was conducted from 27 July to 17 August 2017.

### 2.2. Hyperscanning fMRI System

To measure the neural activation during a conversation between pairs of participants, we used two MRI scanners (Magnetom Verio 3T, Siemens, Erlangen, Germany). The two scanners were used simultaneously along with online video cameras (NAC Image Technology, Yokohama, Japan), MRI-compatible active noise-canceling microphones (Opto ACTIVE II, Kobatel, Yokohama, Japan), and headphones (KIYOHARA-KOUGAKU, Tokyo, Japan) to allow real-time mutual audio-visual communication. Using the noise-canceling microphones, participants could talk to each other during imaging. We used a Siemens Verio 32-channel phased-array coil modified to consist of 24 channels. The standard 32-channel phased-array coil consists of a bottom component comprising 20 channels and a top component of 12 channels. To establish face-to-face communication in this study, it was important to ensure that the top part of the apparatus did not cover the eye region. To visualize the eye region fully, we replaced the top component of the standard 32-channel coil with a 4-channel small flex coil (Siemens) attached with a special holding fixture (Takashima Seisakusho Co., Tokyo, Japan; see [32], for details)

### 2.3. Stimulus Presentation

The presentation software controlled the visual stimuli for the tasks (Neurobehavioral Systems, Berkeley, CA, USA). Video images of participants’ faces were captured using an online video camera system and combined using a Picture-in-Picture system (NAC Image technology and Panasonic System Solutions Japan Co. Ltd., Tokyo, Japan). The participants could see the visual stimulus in a mirror in front of their faces. An LCD projector (CP-SX 12000J, Hitachi Ltd., Tokyo, Japan) projected the combined visual stimuli onto a half-transparent screen placed on the scanner bed approximately 190.8 cm from the participant’s eyes. The stimuli were presented at a visual angle of 13.06° × 10.45° [32].

### 2.4. MRI Data Acquisition

MRI time series data were acquired using ascending-order T2*-weighted gradient-echo echo-planar imaging (EPI) with a multiband sequence developed at the University of Minnesota (Minneapolis, MN, USA) [33,34,35]. Each volume consisted of 66 slices (2.0 mm thickness and a 0.4 mm gap) covering the entire cerebral cortex and cerebellum. MRI time series data were acquired using multiband EPI sequences and the following conditions: repetition time (TR), 1000 ms; flip angle (FA), 55°; echo time (TE), 30.8 ms; multiband acceleration factor, 6; field-of-view (FOV), 216 mm. The size of the in-plane matrix was 90 × 90 pixels; thus, the size of one pixel was 2.4 × 2.4 mm^2^. We acquired 524 volumes (8 min 44 s) per MRI run. For anatomical reference, high-resolution T1-weighted images were obtained with three-dimensional (3D) magnetization-prepared rapid-acquisition gradient-echo sequences (224 slices; thickness = 0.8 mm; TR = 2400 ms; TI = 1060 ms; TE = 2.24 ms; FA = 8°; FOV = 256 mm; voxel dimensions = 0.8 × 0.8 × 0.8 mm^2^).

### 2.5. Task Design

#### 2.5.1. Task Conditions

The experimental task included three factors. The first factor was the type of target to be evaluated and described by partners. There were two types of targets: “introspection” (preference and familiarity), which involved evaluating one’s self-attitude toward the target words; and “object feature identification”, which involved identifying the features of target objects. The second factor was the role of the partner based on their speaking order. There were two roles: “initiator” (the partner who spoke first) and “responder” (the partner who spoke after the initiator). The third factor was a matching condition based on whether the two participants’ disclosures were the same or different. For the matching condition, we checked the participants’ actual responses after the experiment and assigned them to the “match” or “mismatch” conditions.

#### 2.5.2. Introspection (Preference/Familiarity) Tasks

During the introspection tasks, the participants evaluated their preference for or familiarity with the presented items, which they disclosed to their partner in a two-alternative forced-choice manner (like/dislike, familiar/unfamiliar). The task had two phases: evaluation and disclosure. The evaluation phase was regarded as a preparatory phase for the following utterance for disclosure, starting from the initiator’s specification of the task (preference/familiarity for introspection task, color/shape for object identification) followed by the responder’s response “Roger”. Specifically, inside the scanner, paired participants viewed their partner’s face on the screen for 2 s. A colored rectangle frame surrounded the partner’s face for 2 s, signaling the participant’s role. Cyan or magenta frames indicated whether the participant was the initiator or responder, respectively. Below the colored frame, a target word appeared describing everyday items (food, location, animal, commodities, etc.; Figure 1). Simultaneously, the cue (“preference” or “familiarity”) was presented on the initiator’s screen and “XXX” was presented as a placeholder on the responder’s screen (the number of characters was equal to the number of letters in the cue; Figure 1). When the frame and the cue disappeared, the initiator uttered their “preference” or “familiarity” to the responder, who replied with “roger” and considered the answer provided by the initiator. This cue and reply period lasted for 3 s. Next, the disclosure phase started with the colored frames appearing again for 2 s. The order may have differed from that in the previous evaluation phase (first role). When the frames disappeared, the initiator disclosed their opinion (e.g., “I like it” or “I dislike it”/“It’s familiar” or “It’s unfamiliar”), and the responder replied with their opinion. If the responder agreed with the initiator, he/she replied, “I (dis-) like it too” or “It’s (un-) familiar to me too”. If the responder disagreed with the initiator, he/she replied, “I (dis-) like it” or “It’s (un-) familiar to me”. The answer and reply period lasted for 4 s. Thus, an introspection task took 13 s. Throughout the fMRI session, the introspection condition consisted of 96 preference trials and 48 familiarity trials.

To adjust the number of mismatch and match trials, different words were presented to the paired participants in half of the introspection task trials. One participant was presented a target word with a high percentage of “like” responses, and the other was presented with a target word with a high percentage of “dislike” responses. The answers naturally conflicted in the expected-to-mismatch trials; in the mismatch trials, participants were not informed that they had been presented with different words. We confirmed that no one noticed this control at the debriefing. Target words with a high percentage (≥95%) of (dis-) like responses were used in the expected-to-match trials, whereas target words with slightly higher (dis-) like responses (>70%) were used in the expected-to-mismatch trials. In the expected-to-mismatch trials, one participant was exposed to a word that >70% of people liked. In contrast, the other participant was given a word that >70% of people disliked, creating a situation wherein it was typical to have different preferences. With this protocol, the rate of matched responses was approximately 60% during the evaluation of introspection.

Similarly, in the familiarity condition, pairs of participations were presented with the same target words in the expected-to-match trial and a different target word in the expected-to-mismatch trial.

#### 2.5.3. Object Feature Identification Task

The object feature identification tasks were identical to the introspection tasks except for the instruction cue and the target object presentation. Instead of the target word, a picture of the target object appeared on the screen in the role and stimulus periods (Figure 1). The object had two features: shape (star, heart, circle, or square) and color (red, blue, yellow, or green). The cue (“shape” or “color”) appeared on the initiator’s screen, whereas “XXX” was presented as a placeholder on the responder’s screen (the number of characters was equal to the number of letters in the cue; the role and stimulus periods lasted for 2 s each). When the frames and cues disappeared, the initiator uttered the “shape” or “color” to the responder. The responder attended to those features of the object and replied with “roger”. The reply period lasted for 3 s. During the answer and reply period, the initiator stated the feature of the component (“It’s yellow” or “it’s a heart”), to which the responder replied (e.g., “I also see yellow [a heart]” or “I see blue [a star]”). We presented the same object on the two participants’ screens in the matched trial and different objects in the mismatch trial to achieve a 50% matched response rate for the feature evaluation task. Before the experiment, we explained to the participants that different objects might be shown in the object feature identification task, leading to conflicting answers. Throughout the fMRI session, the object feature identification condition consisted of 48 trials.

#### 2.5.4. Adjustment of the Preference/Familiarity of Target Words

To adjust the preference of the target words, we first selected test words based on an online survey before the fMRI experiment. Participants were recruited for the web-based survey through CrowdWorks (CrowdWorks, Inc., Tokyo, Japan). Each participant was paid 100 yen for participating in the survey. A total of 400 volunteers participated (107 males, 252 females; 359 valid responses; age range, 18–40 years; mean age, 29.7 ± 5.7 years). We chose 340 well-known words from the NTT Psycholinguistic Database, “Lexical Properties of Japanese”. Participants were asked to respond with “like”, “dislike”, or “unknown” to the presented words. We sorted all words by the percentage of participants who answered “like”. We then adopted 24 words with the highest percentage of “like” responses and 24 words with the lowest percentage of “like” responses for the expected-to-preference match trials. Moreover, we adopted 24 words with >70% “like” responses and 24 words with <30% “like” responses for the preference expected-to-mismatch trials (see Task design Section 2.5.2. for details). Thus, we selected 96 words for preference conditions. The stimuli for the familiarity condition were selected from those used in the survey, and the familiarity was determined by the experimenter. We defined “familiar” as things that are commonly encountered in daily life (e.g., television and coffee) and “unfamiliar” as things that are relatively uncommon (e.g., ruins and penguins). For each familiar expected-to-match and mismatch trial, 12 words were selected as familiar and 12 as unfamiliar. Therefore, we selected a total of 48 words for familiarity conditions. We conducted a post hoc evaluation of the familiarity of the 48 words used in the experiment through an online survey among the general population (N = 24). For each word, we calculated the percentage of respondents who were familiar or unfamiliar with the word. Among the words used in the familiarity expected-to-match trial, >75% of the participants answered “familiar” for all 12 words selected as familiar and >75% answered “unfamiliar” for all but 2 of the 12 words selected as unfamiliar. For the words used in the familiarity expected-to-mismatch trial, >75% of the participants answered “familiar” for all 12 words used in the familiarity expected-to-mismatch trial, and >75% answered “unfamiliar” for all but five of the 12 words selected as unfamiliar.

### 2.6. fMRI Imaging Protocol

The fMRI study consisted of six repeated runs. Each fMRI run started with a 10-s rest period and included 16 preference condition trials, eight familiarity condition trials, and eight object identity trials and ended with a 10-s rest period. In a single run, the order of conditions presented was pseudo-randomized by a genetic algorithm [36] that maximized the estimation efficiency for the tested contrasts. We used a variable interstimulus interval (ISI) jitter, and the trials were separated by an ISI of 0–9 s. Brain activity recorded during the 5 s from the role and stimulus presentation through the cue and reply period was analyzed as the evaluation phase (Figure 1, red border). Brain activity recorded during the 6 s from the second role presentation through the answer and reply period was analyzed as the disclosure phase (Figure 1, blue border).

### 2.7. Data Analysis

#### 2.7.1. Image Preprocessing

We performed image preprocessing and statistical analysis using the Statistical Parametric Mapping (SPM12) revision 7487 (Wellcome Centre for Human Neuroimaging, London, UK) implemented in MATLAB 2015a (Mathworks, Natick, MA, USA). First, each participant’s T1-weighted anatomical image was co-registered with the image averaged over 305 T1 images in SPM12. Each co-registered T1-weighted anatomical image was segmented into tissue class images. Next, we realigned functional images from each run to the first image to correct for subjects’ head motion. The segmented T1-weighted anatomical image was then co-registered to the mean of all realigned images. Each co-registered T1-weighted anatomical image was normalized to the MNI space with the DARTEL procedure [37]. Each anatomical image was segmented into tissue class images using a unified approach [38]. The gray and white matter images were registered and normalized to MNI space using a preexisting template based on the brains of 512 Japanese people scanned at the NIPS. DARTEL registration and normalization parameters were applied to each functional image and the T1-weighted anatomical image. The final resolution of the normalized EPI images was 2 × 2 × 2 mm^3^. The normalized functional images were filtered using a Gaussian kernel (full width at half maximum, 8 mm) in the x-, y-, and z-axes.

#### 2.7.2. Statistical Analysis

We adopted a summary statistics approach to depict the neural substrates of the task-related brain activity. In individual analyses, we fitted a general linear model to the fMRI data of each participant. We modeled the neural activity with delta functions convolved with a canonical hemodynamic response function. The design matrix included six regressors specifying six sub-conditions: preference initiator, preference responder, familiarity initiator, familiarity responder, object feature initiator, and object feature responder (Table 1) in the evaluation phase. Similarly, the 12 regressors for the disclosure phase specified 12 sub-conditions: match preference initiator, match preference responder, match familiarity initiator, match familiarity responder, match object feature initiator, match object feature responder, mismatch preference initiator, mismatch preference responder, mismatch familiarity initiator, mismatch familiarity responder, mismatch object feature initiator, and mismatch object feature responder (Table 1). Missed trials in which participants did not respond or missed the stimulus were modeled as regressors of no interest. The onset of the evaluation phase was at the presentation of the first role frame. The duration of one event was 5 s (Figure 1, red border). The onset of the disclosure phase was at the presentation of the second role frame, and each event lasted for 6 s (Figure 1, blue border).

Since it is not possible to completely control the responses of participants, the regressors were modeled by checking the actual responses after the experiment. Because the sample sizes corresponding to each sub-condition per run were small and uneven, all six runs were concatenated and treated as one run. We prepared six regressors corresponding to a high-pass filter per run and five regressors to remove the run-related component. The created regressors were specified as nuisance regressors. The BOLD signal of the frontal region was weak. Therefore, we lowered the masking threshold to 0.1 and excluded any activation outside the gray matter with the explicit mask. No global scaling was performed. Serial temporal autocorrelation of the pooled voxels was estimated with a first-order autoregressive model using the restricted maximum likelihood procedure. The obtained covariance matrix was used to whiten the data [39]. The estimated parameters were calculated using the least-squares estimation on the high-pass-filtered and whitened data and design matrix. The parameter estimates in the individual analyses consisted of contrast images used for the group-level analysis.

We conducted group-level analyses with contrast images of the evaluation phase, resulting in 3 (preference, familiarity, and object feature) × 2 (initiator and responder) factors and the predefined contrasts (Table 1). We also conducted group-level analyses with the contrast images of the evaluation phase, resulting in 3 (preference, familiarity, and object feature) × 2 (initiator and responder) × 2 (match and mismatch) factors and the predefined contrasts (Table 1).

The resulting set of voxel values for each contrast constituted a statistical parametric map of the t statistic (SPM{t}). The statistical threshold was set at *p* < 0.05 with family-wise error (FWE) correction at the cluster level for the entire brain [40] and a height threshold of *p* < 0.001 [41]. We used the *Atlas of the Human Brain*, 4th edition, for anatomical labeling [42].

#### 2.7.3. Generalized PPI Analysis

Contrasts of task-related activation during the evaluation phase showed that the left IFG was part of the networks representing the internal model of introspection (Figure 2). This area also showed introspection-specific mismatch-related activation during the disclosure phase (Figure 3, magenta region). Thus, we hypothesized that the left IFG receives bottom-up information related to introspection for the comparison, probably from the auditory areas (where verbally exchanged information is first cortically received). We performed a generalized psychophysiological interaction (gPPI) analysis [43,44] using the CONN toolbox (version 19. c; [45]) and performed conventional seed-to-voxel gPPI analysis with the whole brain as the search area. The components associated with a linear trend, cerebrospinal fluid, white matter, and experimental tasks (mistrial effects) were removed from the BOLD time series as confounding signals. To denoise the functional images, we used band-pass filtering (0.008–0.09 Hz) and linear detrending filtering but not despiking. The seed region was functionally defined as the cluster in the left IFG that showed significant activation in contrasts under the introspection-related and object feature identification-related mismatch conditions (Figure 3, magenta region). Using the residual time series data, a gPPI analysis was performed to evaluate whether the effective connectivity from the seed region was modulated by the 12 task conditions—that is, the disclosure phase conditions of 2 (match, mismatch) × 3 (preferences, familiarity, object) × 2 (initiator, responder) combinations at the individual level. This individual-level analysis produced contrast images representing the modulation of effective connectivity from the seed region. Task-related activation and PPI were independently analyzed because the left IFG was defined by the task-related activation, whereas PPI was evaluated by correlational analysis of the residual time series data of the left IFG with the rest of the brain. Up to this point, all procedures were conducted using the CONN toolbox.

Finally, we used these contrast images and the random-effects model implemented in SPM12 with the full factorial models used in task-related activation analyses. To depict enhanced effective connectivity by introspection (as compared with object feature identification), we used the contrast of conjunction analysis #11 (#7 and #8) (Table 1) under the assumption that the introspection-specific bottom-up signal was sent irrespective of match or mismatch conditions.

Results were assessed at a significance level of *p* < 0.05 with FWE correction at the cluster level. The height threshold to form each cluster was set at an uncorrected *p* < 0.001 [41,46].

## 3. Results

### 3.1. Behavioral Results

The rates of matching responses were 61.2 ± 5.9% (mean ± SD, N = 19 pairs) for preference, 57.3 ± 5.8% for familiarity, and 50.1 ± 1.1% for object feature identification. A repeated-measures analysis of variance showed a significant main effect of condition (F(2, 36) = 27.34, *p* < 0.001). Post-hoc tests with Bonferroni correction showed that matching rates were significantly higher in the preference and familiarity conditions than in object feature identification (*p* < 0.001).

### 3.2. Functional Imaging Results

The fMRI results showed that the introspection evaluation phase—including both preference and familiarity judgments—contrasted with the object feature identification phase (contrast of conjunction analysis #3 [#1 and #2] in Table 1) and activated the following areas: the dorso-medial prefrontal cortex (dmPFC), extending to the anterior cingulate cortex (ACC), the precuneus, left hippocampus, left temporoparietal junction (TPJ), right cerebellum, bilateral occipital gyrus, left middle frontal gyrus, left inferior temporal gyrus, and left IFG (Figure 2; Table 2).

The introspection disclosure phase contrasted with the object feature identification phase (Contrast #11) and activated regions similar to those activated during the evaluation phase. Irrespective of whether the participant’s statements were matched or mismatched, the dmPFC (extending to the ACC), precuneus, left hippocampus, left TPJ, bilateral temporal pole, and right cerebellum were activated. Unlike in the evaluation phase, the basal nucleus was also activated (Figure 3, cyan region; Table 3). When the two participants’ statements were mismatched (Contrast #12), the left IFG was activated more than when they were matched (Figure 3, magenta region; Table 3).

The PPI analysis with the left IFG as the seed region revealed that introspection enhanced FC in the bilateral superior temporal gyrus (STG) and bilateral precentral gyrus (Figure 4, green region; Table 4). These regions in the dorsal portion of the STG ventrally overlapped with those that showed the mismatch detection-related activity that was common to introspection and object feature identification (Contrast #13, Figure 4, red region; Table 4).

## 4. Discussion

During the conversation in which paired participants disclosed their introspection regarding preference for and familiarity with a presented object, the left IFG showed introspection-specific mismatch-related activity during the disclosure phase on real-time hyperscanning fMRI. This novel finding was consistent with our hypothesis and indicates the role of the left IFG in the alignment of the conversation regarding interpersonal comparison.

### 4.1. Introspection-Related Activation of the Evaluation Phase

A preference for and familiarity with presented objects was defined as introspection that can be compared verbally and interpersonally. During the introspection of the object in terms of preference or familiarity, we confirmed the activation of the left IFG and the mentalizing network, including the dmPFC, posterior cingulate cortex (PCC), left TPJ, and the right cerebellum. The observed activation of the mentalizing network is consistent with a previous meta-analysis of self- and other judgments [47] that revealed that self-related judgments involve left-lateralized activity. Mentalizing networks—especially the medial prefrontal cortex (mPFC), PCC, and TPJ—are activated by introspection to identify self-traits [48,49]. van Overwalle [49] suggested that trait information about unfamiliar others selectively engages the dorsal part of the mPFC, whereas the ventral part is implicated when making trait inferences about familiar others or the self. Dynamic causal modeling applied to fMRI data obtained during introspection tasks revealed that self-related processes were driven by the PCC and modulated by the mPFC [48].

The neural substrates of introspection of emotion and thoughts are known to overlap with those of unconstrained thought (default mode network), particularly the dorso-medial prefrontal and the medial parietal cortex [19]. The default mode network (DMN) consists of a set of interacting hubs and subsystems, including the medial temporal lobe and dmPFC subsystems and the anterior medial prefrontal and PCC hubs. In particular, the dmPFC subsystem is related to introspection about the mental states of the self and others (reviewed in [20]). Recent studies have explored the close relationship between the mentalizing network and the cerebellum (reviewed in [50]). The cerebellum, especially the posterior cerebellum (Crus I and II), contributes to mentalizing and has a reciprocal connection with the TPJ. The functional role of the cerebellum is postulated to be the prediction of social action sequences (social sequencing hypothesis) [51]. The left IFG is not part of the mentalizing network or the DMN but is associated with semantic memory retrieval [52,53], social knowledge [54], personality traits [55], and inner speech [56]. Thus, the left IFG is a vital region functionally located at the intersection of language and social roles [57]. For this reason, it is conceivable that the mental model of introspection was represented by the mentalizing network in conjunction with the cerebellum and the left IFG.

### 4.2. Introspection-Specific Mismatch-Specific Activity in the Left IFG during the Disclosure Phase

In the present study, the disclosure phase was associated with introspection-specific mismatch-specific activity (compared with the match condition) in the orbital portion of the left IFG (BA 47). However, this pattern was not observed in the mentalizing network (including the mPFC). These findings suggested functional differentiation within the neural network, as depicted during the evaluation phase.

In terms of linguistic processing, the left IFG exhibits a functional gradient across the dorso-ventral axis and includes the following components: phonologic, syntactic, and semantic [58]. A previous study reported that the ventral portion of the left IFG—corresponding to BA 47 (and part of BA 45)—shows mismatch detection-related activity specific to preference/familiarity. As BA 47 is related to the semantic process [59], the mismatch detection-related activity likely represents the comparison between the forward model of introspection of the self and the partner. This interpretation is consistent with that of previous studies on self–other processes that utilized language as a semantic task [60,61,62]. Kelley et al. [60] addressed social and linguistic demands in self–other processing using the social observation approach. Participants were asked to judge whether the adjectives presented described themselves (self-relevance) or their favorite teacher (other-relevance) or to judge the case of the presented words (control). Compared with case judgments, the relevance judgments were accompanied by activation of the left IFG and ACC. Additionally, a separate region of the mPFC was selectively engaged during self-referential processing. The authors concluded that self-referential processing could be functionally dissociated from other forms of semantic processing within the human brain. However, it was unclear how the self-referencing and semantic processes interact during social interaction.

Previous studies adopted the inference of a third person’s mental state or personal trait in their experimental paradigm. In contrast, the present study included an exchange based on the evaluation of shared targets within a real-time conversational context, which required an introspective process. The left IFG showed introspection-specific activation during the evaluation phase, indicating that it is part of the internal introspection model. According to the hierarchical predictive coding scheme, the bottom-up signal may originate from the lower representation (which shows the introspection-non-specific mismatch-related activity).

We found that the bilateral STG/superior temporal sulcus and right IPL showed mismatch detection-related activity for both introspection and object feature identification, indicating that these areas are responsible for processing the match/mismatch of the utterances. PPI analysis using the left IFG as the seed revealed that compared with object feature identification, the introspection disclosure phase showed enhanced FC with the bilateral STG and the bilateral primary motor cortex under both match and mismatch conditions. The bilateral STG close to the superior temporal sulci showed partial overlap with the areas responsible for the match/mismatch of the utterances. Previous fMRI [63] and electrophysiological [64] studies have reported human voice-selective responses along the upper bank of the superior temporal sulcus. The bilateral superior temporal sulcus and the left IFG are known to comprise the network for speech perception (that is, sublexical processing) (reviewed in [27]). Furthermore, the IFG (the core of the MNS) receives information from the extended motor MNS—including the insula, middle temporal gyrus, posterior part of the superior temporal sulcus, dorsal part of the premotor cortex, and the primary sensorimotor cortex—to critically transform the information essential for the motor simulation outcome [26]. The enhanced connectivity with the bilateral primary motor areas may represent the bottom-up transfer of self-utterance or the auditory mirror responses [65]. Thus, during the introspection disclosure phase, the bilateral STG and the primary motor cortex likely provided lower-level information regarding speech perception to the left IFG so as to generate the mismatch-related activity in terms of the introspection of the self and others. We did not observe the enhanced FC of the left IFG with the mentalizing network, the latter of which represents self-introspection. This finding is consistent with the fact that the present task did not require the updating of the self-introspection.

Additionally, the bilateral temporal pole and the basal nucleus were activated during the introspection disclosure. The temporal pole is implicated in personal identity [66], and it constitutes a subsystem of the DMN along with the dmPFC, TPJ, and lateral temporal cortex [20]. The basal forebrain, including the basal nucleus, is associated with driving changes in DMN activity during rest-to-task transitions [67]. Thus, activation in the basal forebrain may be involved in the attentional shift from introspection during the evaluation phase toward self–other comparisons during the disclosure phase.

### 4.3. Comparison with a Previous Study by Yoshioka et al. [2]

Yoshioka et al. [2] used hyperscanning fMRI to show that the sharing of the mental model of the state of a conversation is represented by the interindividual neural synchronization of the mentalizing network. This provided evidence for the shared mental model. However, the between-individual mismatch was not tested by Yoshioka et al. [2], whose experimental design did not include the “mismatch” condition in which participant’s forward model of the object and the partner’s utterance are discordant. Another difference was the information being shared. Yoshioka et al. [2] found that the visual experience (“I see this flower is yellow”) was neurally represented by the interindividual neural synchronization of the DMN. However, the researchers did not test whether more subjective experiences, such as introspection, can be exchanged. Verbal communication enables us to exchange our attitudes toward the shared object as follows: “I know I like this flower” or “I know I am familiar with this tool”. These propositions are representative contents describing one’s evaluative attitude toward the object. We regard this appreciation of self-attitude as an example of introspection, defined as the capacity to attend to one’s thoughts and feelings [19]. The present study showed that when comparing introspection through conversation, the mentalizing network and the auditory MNS generated hierarchical predictive coding structures in which the hub is the left IFG characterized by the introspection-specific mismatch-related activation.

### 4.4. Limitations and Future Perspective

The present study has several limitations. First, the words used for familiarity conditions were selected differently from those used for preference conditions. We originally inferred that the familiarity decision requires introspecting personal experiences and life circumstances, and is therefore more difficult to control than the preference decision. We tried to control for this as much as possible by selecting words for familiarity conditions based on the subjective evaluation of the experimenter regarding the popularity and daily usage of the words. We defined familiar objects as things that are commonly seen (e.g., television, coffee) and unfamiliar objects as things that are uncommonly seen (e.g., ruins, penguins) in daily life. Since the matching rate for the familiarity condition was 57% in the fMRI experiment (with no significant differences with the preference condition), the word selection procedure has no effect on the conclusion of the present study per se. To confirm the relevance of the subjective evaluation, we conducted a post-hoc evaluation of the familiarity rates of the stimuli used in the experiment.

Second, in the familiarity task, the number of fMRI data samples for the mismatch conditions was smaller than that for the object feature identification task. This difference could not be controlled completely, partly because preference and familiarity varied across individuals and partly because the age of participants differed between the fMRI study and the stimulus preparation experiment. However, in the imaging analysis, the difference in the number of conditions corresponds to the number of epochs of each condition at individual-level analysis. Because the height of each epoch is constant in each model vector, the epoch number does not affect the beta estimates of each vector, which were incorporated into the second-level analysis.

Third, the sample size was relatively small (N = 38, 19 pairs of participants), although it was comparable with those of a previous hyperscanning fMRI study with similar settings ([2]; N = 44). Nevertheless, we identified the neural substrates for self–other comparisons of introspection within a social interaction context.

Fourth, our task did not completely control the presented objects between the introspection tasks and identification tasks: the former adopted words naming the object, whereas the latter adopted a visual object with the target feature words (such as color and shape). As the introspection tasks include object identification because familiarity or preference judgment occurs when the target object is identified, the contrast between introspection and object feature identification (Contrasts 1, 7, and 8 in Table 1) includes the introspection-related activity, with more/less verbal (motor) production differences. On the other hand, Contrast 9 compared the Mismatch-Match difference of introspection with that of object feature identification (Table 1). This contrast controlled for the verbal component within introspection and object feature identification. Thus Contrast 12 (=conjunction of contrast 4 and 9) depicted the introspection-specific mismatch-specific activity, which was seen in the left IFG (Figure 3).

Fifth, the present study utilized verbal communication dealing with introspection without any control involving a non-verbal setup. Therefore, the role of the left IFG cannot be generalized for the mismatch detection in introspection or metacognition [68]. Thus, future studies are warranted.

Sixth, the FC within the left IFG was not evaluated. The present study demonstrated introspection-specific mismatch activity in the orbital portion of the left IFG (BA 47). According to the dual-stream model [29], the dorsal stream targets BA 44, whereas the ventral stream ends in BA 45 (reviewed in [30]). Future studies are warranted to reveal the convergent patterns of the ventral and dorsal streams with the BA 47 in the IFG.

Finally, we included healthy adults only. Applying the current experimental design to participants who experience psychiatric disorders—such as autism spectrum disorder (ASD)—would be of clinical interest. ASD is characterized by impairments in social communication and restricted, repetitive patterns of behaviors, interests, or activities [69]. The former manifestation (ASD) is related to difficulty in reciprocal social interaction, stemming from an atypical representation of the self and others [57]. Utilizing verbal self- and other-referencing tasks, Kana et al. [57] reported the hypoactivation of the left IFG and IPL in the ASD group and concluded that this hypoactivation may represent reduced semantic and social processing. Kana et al. [57] utilized the “single-brain, third-person” approach; as such, the “second-person” approach adopted in the present study could provide additional information regarding the self–other comparisons of individuals with ASD during social interaction.

## 5. Conclusions

The neural substrates involved in exchanging information regarding introspection during conversation consist of the mentalizing network and the left IFG. The left IFG integrates the bottom-up information related to others from the auditory MNS, thus constituting the neural representation of alignment.

## Figures and Tables

**Figure 1 brainsci-13-00111-f001:**
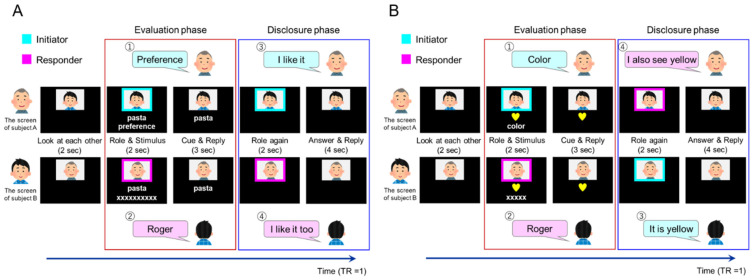
Time course of one trial in the (**A**) introspection task and (**B**) object feature identification task. The number in the upper left corner of the balloon indicates an example of the order of utterance. Brain activity was analyzed in the evaluation (red border) and disclosure (blue border) phases.

**Figure 2 brainsci-13-00111-f002:**
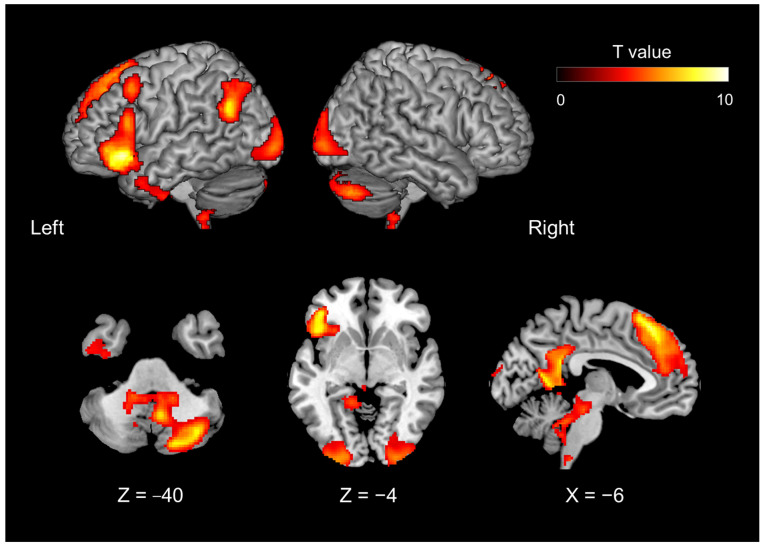
Brain activity during the evaluation phase of introspection (including both preference and familiarity judgments), contrasted with object feature identification (Contrast #3, Table 1).

**Figure 3 brainsci-13-00111-f003:**
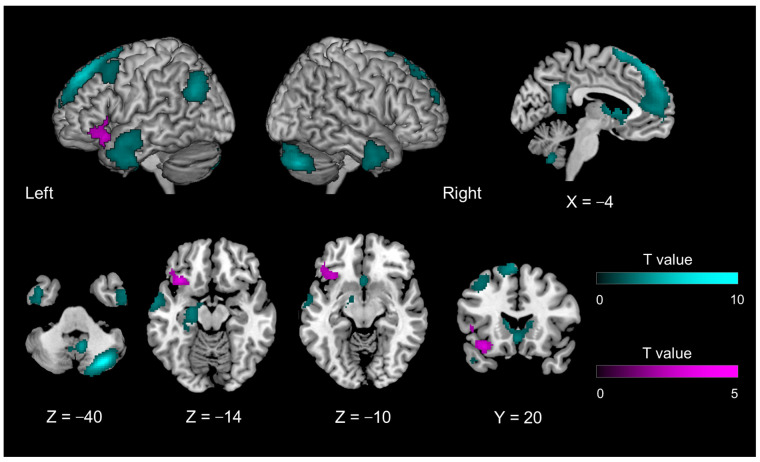
Brain activity during the disclosure phase of introspection relative to object feature identification, irrespective of mismatch and match conditions (Contrast #11, Table 1) (cyan region); and introspection-specific mismatch as compared with object feature identification-related mismatch (Contrast #12, Table 1) (magenta region).

**Figure 4 brainsci-13-00111-f004:**
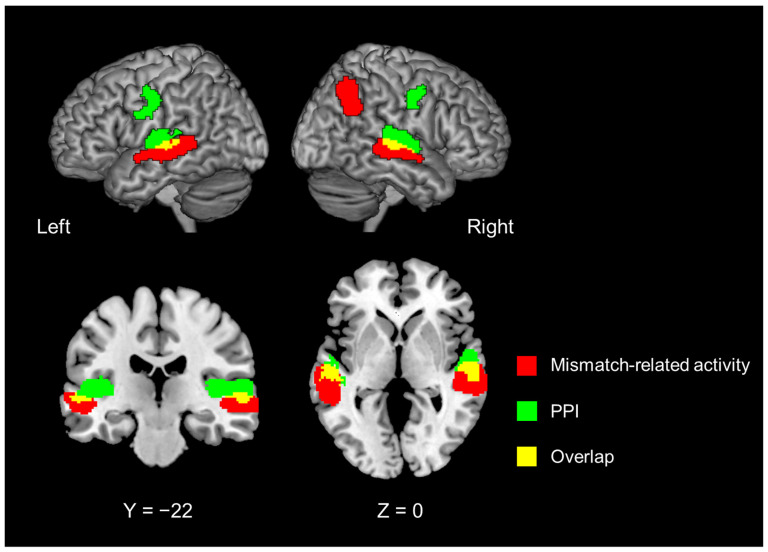
Mismatch-related brain activity, irrespective of introspection or object feature identification (Contrast #13, Table 1) (red region). Significant regions identified with psychophysiological interaction analysis with the left inferior frontal gyrus as the seed region (green region). Areas shown in yellow indicate an overlap between the two (red and green) parts.

**Table 1 brainsci-13-00111-t001:** Predefined contrasts of the evaluation phase and disclosure phase conditions.

Evaluation Phase													
contrast #		Preference				Familiarity				Object feature			
		I		R		I		R		I		R	
1	Introspection > Object feature identification	1		1		1		1		−2		−2	
2	Introspection > 0	1		1		1		1		0		0	
Conjunction analysis													
3	#1 & #2	Introspection-specific activity compared with object feature identification											
**Disclosure phase**													
		Match						Mismatch					
		Preference		Familiarity		Object feature		Preference		Familiarity		Object feature	
contrast #		I	R	I	R	I	R	I	R	I	R	I	R
4	Mismatch introspection > 0	0	0	0	0	0	0	1	1	1	1	0	0
5	Mismatch object feature identification > 0	0	0	0	0	0	0	0	0	0	0	1	1
6	Mismatch introspection > Match introspection	−1	−1	−1	−1	0	0	1	1	1	1	0	0
7	Match (introspection > object feature identification)	1	1	1	1	−2	−2	0	0	0	0	0	0
8	Mismatch (introspection > object feature identification)	0	0	0	0	0	0	1	1	1	1	−2	−2
9	(Mismatch-Match) introspection > (Mismatch-Match) object feature identification	−1	−1	−1	−1	2	2	1	1	1	1	−2	−2
10	Mismatch object feature identification > Match object feature identification	0	0	0	0	−1	−1	0	0	0	0	1	1
Conjunction analysis													
11	#7 & #8	Introspection relative to object feature identification, irrespective of mismatch and match.											
12	#4 & #9	Introspection-specific mismatch compared with object feature identification-related mismatch.											
13	#4 & #5 & #6 & #10	Mismatch-related activity, irrespective of introspection or object feature identification.											

**Table 2 brainsci-13-00111-t002:** Significant clusters of brain activity in the evaluation phase of introspection, including both preference and familiarity judgments, as contrasted with object feature identification. The locations of local maxima were defined according to Mai et al. [42].

Cluster Size	MNI-Coordinates of Peak-Voxel			*p*-Value	*t*-Value	Hemisphere	Location
(mm^3^)	x	y	z	(FWE-corr)			
16,256	−44	26	−8	<0.0001	10.691	left	inferior frontal gyrus, orbital part
	−30	22	−6		4.878	left	basal operculum
8488	−46	−58	30	0.0001	9.034	left	superior temporal gyrus
	−40	−70	44		5.459	left	angular gyrus
21,576	−8	32	50	<0.0001	8.788	left	superior frontal gyrus, medial part
	−16	42	42		6.303	left	superior frontal gyrus, lateral part
	−6	42	22		5.387	left	cingulate sulcus, anterior part
47,944	16	−80	−30	<0.0001	8.125	right	cerebellum CrusI
	6	−58	−42		8.077	right	cerebellum IX
	−14	−48	2		8.007	left	occipital gyrus
	−10	−56	6		7.675	left	calcarine salcus, anterior part
	28	−76	−42		7.673	right	cerebellum CrusII
	−4	−60	10		7.509	left	precuneus
	−6	−46	22		7.342	left	splenium of corpus callosum
	16	−44	4		6.505	right	hippocampal fissure
	−2	−24	−18		6.008	left	pons
	−8	−46	−34		5.930	left	cerebellum IX
	−6	−46	8		5.898	left	isthmus of cingulate gyrus
	−18	−40	−8		5.536	left	perisplenial region
	16	−42	−36		5.477	right	cerebellum X
	−26	−32	−22		5.251	left	parahippocampal gyrus
	−16	−42	−40		5.243	left	cerebellum X
	−6	−40	−28		5.099	left	cerebellum I-IV
	−22	−18	−26		5.030	left	cingulum (hippocampal part)
	14	−42	−60		4.801	right	cerebellum VIIIb
	−44	−8	−44		4.762	left	fusiform gyrus
	−22	−48	−52		4.590	left	cerebellum VIIIb
	−16	−22	−14		4.491	left	cerebral peduncle
	−20	−52	−40		4.449	left	cerebellum Dentate
	−44	8	−34		4.432	left	inferior temporal gyrus
	2	−34	−18		4.304	right	decussation of the superior cerebellar peduncle
	26	−48	8		3.909	right	CA3 field of hippocampus
	−4	−30	−6		3.575	left	central tegmental tract
2720	−38	18	44	0.0370	6.649	left	middle frontal gyrus
7288	−26	−100	−2	0.0003	6.007	left	inferior occipital gyrus
	−22	−102	10		5.912	left	superior occipital gyrus, lat. part
	−24	−82	−6		4.473	left	inferior lingual gyrus, lateral part
	−14	−98	16		3.964	left	gyrus descendens
6168	26	−98	0	0.0009	5.933	right	inferior occipital gyrus
	26	−98	18		5.124	right	superior occipital gyrus, lat. part

**Table 3 brainsci-13-00111-t003:** Significant clusters of brain activity of introspection relative to object feature identification irrespective of mismatch and match conditions (Figure 3, cyan region) and of introspection-specific mismatch as compared with object feature identification-related mismatch (Figure 3, magenta region). The locations of local maxima were defined according to Mai et al. [42].

Cluster Size	MNI-Coordinates of Peak-Voxel			*p*-Value	*t*-Value	Hemisphere	Location
(mm^3^)	x	y	z	(FWE-corr)			
The brain activity of the introspection-relative to object feature identification, irrespective of mismatch and match							
13,656	30	−74	−36	<0.0001	11.953	right	cerebellum CrusI
33,168	−14	46	44	<0.0001	8.776	left	superior frontal gyrus, lateral part
	−2	56	26		6.649	left	superior frontal gyrus, medial part
	−2	58	−6		5.275	left	inferior frontopolar gyrus
	4	46	48		4.776	right	superior frontal gyrus, medial part
	−40	16	52		4.511	left	middle frontal gyrus
	6	40	56		3.872	right	superior frontal gyrus, lateral part
	−4	68	6		3.260	left	middle frontopolar gyrus
3712	4	−60	−48	0.0115	8.303	right	cerebellum IX
6048	−52	−68	38	0.0010	6.936	left	angular gyrus
6648	−4	−52	26	0.0006	6.086	left	isthmus of cingulate gyrus
9632	−44	0	−36	<0.0001	5.444	left	inferior temporal gyrus
	−58	6	−18		5.235	left	middle temporal gyrus
6432	0	16	−6	0.0007	5.433		cingulate gyrus
	−4	0	6		5.090	left	bed. N. of the stria terminalis, medial div.
	−6	14	6		5.088	left	fundus region of caudate n.
	6	10	2		4.940	right	accumbens n., lateral p.
	12	20	8		3.981	right	medial caudate n.
2488	−22	−14	−16	0.0496	5.143	left	CA3 field of hippocampas
	−12	−4	−10		3.745	left	lateral hypothalamic area
	−28	−30	−14		3.556	left	subiculum
6480	54	0	−36	0.0007	5.116	right	inferior temporal sulcus
	50	−2	−28		4.597	right	middle temporal gyrus
Introspection-specific mismatch compared with object feature identification-related mismatch							
2576	−34	22	−16	0.0444	4.664	left	transverse insular gyrus
	−46	32	−10		3.546	left	inferior frontal gyrus, orbital part
	−48	20	4		3.347	left	inferior frontal gyrus, triangular part

**Table 4 brainsci-13-00111-t004:** Significant clusters of mismatch-related activity irrespective of introspection or object feature identification (Figure 4, red region) and psychophysiological interaction analysis with the left inferior frontal gyrus as the seed region (Figure 4, green region). The locations of local maxima were defined according to Mai et al. [42].

Cluster Size	MNI-Coordinates of Peak-Voxel			*p*-Value	*t*-Value	Hemisphere	Location
(mm^3^)	x	y	z	(FWE-corr)			
The mismatch-related activity, irrespective of introspection or object feature identification.							
8696	58	−18	−2	<0.0001	5.419	right	superior temporal gyrus
8552	−54	−24	−4	0.0001	5.015	left	superior temporal gyrus
4320	50	−52	32	0.0059	4.925	right	superior temporal gyrus
	44	−62	48		4.066	right	angular gyrus
Psychophysiological interaction analysis with the left IFG as the seed region.							
7912	−40	−30	10	<0.0001	5.494	left	ant. transverse temporal gyrus
	−64	−30	4		3.368	left	superior temporal gyrus
8400	52	−14	4	<0.0001	5.367	right	ant. transverse temporal gyrus
	60	−24	10		5.190	right	superior temporal gyrus
	56	−6	−2		4.561	right	planum polare, lat. part
2688	−50	−8	44	0.0002	4.108	left	precentral gyrus
	−38	−16	34		3.553	left	superior longitudinal fascicle III, vent. comp.
1520	50	−4	42	0.0069	4.041	right	precentral gyrus

## Data Availability

The datasets generated in this study are available from the corresponding author upon reasonable request.

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
