# Peer review of "The Role of the Left Inferior Frontal Gyrus in Introspection during Verbal Communication"

_brainsci, 2023, doi:10.3390/brainsci13010111_

Round 1

Reviewer 1 Report

Dear Authors,

I read your work entitled “The role of the left inferior frontal gyrus in introspection during verbal communication” and here I enclose my recommendations to you:

1.     Ι suggest the Authors to check their text for some editing of English and possible language errors.

2.     The Authors must include information is their participants were evaluated for their cognitive, communication and language status. If not, this must be added as a limitation to this study.

Thank you.

Author Response

Reviewer1

R1-1     Ι suggest the Authors to check their text for some editing of English and possible language errors.

Response:           

The entire text has been checked by the authors followed by a final editorial checkup by professional, scientific editors (Editage).

R1-2      The Authors must include information is their participants were evaluated for their cognitive, communication and language status. If not, this must be added as a limitation to this study.

Response:

All participants were native Japanese-speaking people participating in ordinary social interactions such as schooling and housekeeping. This point has now been clarified in section 2.1.  

Reviewer 2 Report

Kindly, see the comments attached.

This study has examined the activation associated with the introspection effects of verbal communication using fMRI hyperscanning and data collected from 19 pairs of healthy participants who completed introspection and object feature identification tasks. The IFG regions and default mode network regions (dmPFC and PCC) were found to be significantly engaged in the task and were suggested to be the key contributors to the task. A seed-based functional connectivity analysis supported bottom-up processes of task activation. While I support this work for publication (I found the findings well presented with compelling details), I have some concerns about the complexity of the task, the novelty of the findings, and the direction of the interpretations. I wonder if the authors can address them at this point. Specific details are provided below:

As a major issue, the task looks very complex for generalization; it involves interactions between multiple people with many subjective decisions. Furthermore, there is no clear hypothesis (or expectation) regarding the task conditions. I'm no expert, but two object identification and introspection tasks are already complex, so they shouldn't be directly compared. Maybe a simple control condition, e.g., fixation, could be better suited to the contrast for each of the task conditions. I came across a study by Kühn et al. (2014) that used inner speech and fixation to investigate the effects of introspection. Nevertheless, it would be helpful to clarify the key advantage of the current design over existing tasks. Also, I had a hard time understanding the difference between the disclosure phase and the evaluation phase. I'm curious if there are any additional details that can be shared, such as a video file of the task completion, as supplementary material.

Due to the task's complexity, a wide range of functional networks could have been involved in the task. The IFG's engagement, in particular, could be influenced more by verbal (motor) communications than by introspection effects. Was there a way to control the speech-related responses? As for the network of activation (of the introspection effects), the engagement of bilateral (anterior) temporal lobe and left angular gyrus regions seems very interesting, as shown in Fig. 3, but there are minimal inferences about those effects. These regions are known to be involved in semantic processes and could be the key selling point of the work, but they have received the least attention. Last but not least, the use of connectivity analysis confirms that the IFG is engaged in auditory and motor sensory regions and is less involved in semantic processes that are relevant to mental states. Perhaps other regions or a cluster of regions could be used as seeds for connectivity to confirm this.

Minor comments,

  • Please specify which brain atlas was used to report regions of interests.

Reference,

Kühn S, Fernyhough C, Alderson-Day B and Hurlburt RT (2014) Inner experience in the scanner: can high fidelity apprehensions of inner experience be integrated with fMRI? Front. Psychol. 5:1393. doi: 10.3389/fpsyg.2014.01393

Author Response

Reviewer 2

Comments and Suggestions for Authors

Kindly, see the comments attached.

This study has examined the activation associated with the introspection effects of verbal communication using fMRI hyperscanning and data collected from 19 pairs of healthy participants who completed introspection and object feature identification tasks. The IFG regions and default mode network regions (dmPFC and PCC) were found to be significantly engaged in the task and were suggested to be the key contributors to the task. A seed-based functional connectivity analysis supported bottom-up processes of task activation. While I support this work for publication (I found the findings well presented with compelling details), I have some concerns about the complexity of the task, the novelty of the findings, and the direction of the interpretations. I wonder if the authors can address them at this point. Specific details are provided below:

R2-1     As a major issue, the task looks very complex for generalization; it involves interactions between multiple people with many subjective decisions. Furthermore, there is no clear hypothesis (or expectation) regarding the task conditions. I'm no expert, but two object identification and introspection tasks are already complex, so they shouldn't be directly compared. Maybe a simple control condition, e.g., fixation, could be better suited to the contrast for each of the task conditions. I came across a study by Kühn et al. (2014) that used inner speech and fixation to investigate the effects of introspection. Nevertheless, it would be helpful to clarify the key advantage of the current design over existing tasks. Also, I had a hard time understanding the difference between the disclosure phase and the evaluation phase. I'm curious if there are any additional details that can be shared, such as a video file of the task completion, as supplementary material.

Response:

As the reviewer suggested, the task is relatively complex because of dealing with the interaction between paired participants. The contrast between introspection and object identification is based on the premise that the introspection tasks include object identification because familiarity or preference judgment occurs when the target object is identified. Thus, our hypothesis was that the contrast between introspection and object identification depicts introspection-specific activity.

As the reviewer pointed out, our task did not completely control the presented objects between the introspection tasks and identification task: the former adopted words naming the object, whereas the latter adopted a visual object with target feature words (such as color and shape). This incomplete control issue is now mentioned in the limitation section (4.4.) as follows:

“Fourth, our task did not completely control the presented objects between the introspection tasks and identification tasks: the former adopted words naming the object, whereas the latter adopted a visual object with target feature words (such as color and shape). The introspection tasks include object identification because familiarity or preference judgment occurs when the target object is identified. Thus, the contrast between introspection and object identification depicts the introspection-specific activity.”

The key advantage of the present task was that this design allows the clarification of the inner experience, introspection during the evaluation phase, which should be disclosed during the following disclosure phase. The inner experience is known to be mediated by inner speaking (Kuhn et al. 2014) [1], which component should be subtracted out by the contrast between the introspection and object identification tasks. This point is now discussed in the Limitation section (4.4).

The evaluation phase is regarded as a preparatory phase for the following utterance for disclosure, starting from the initiator’s specification of the task (preference/familiarity for introspection task, color/shape for object identification) followed by the responder’s response “Roger.” During the disclosure phase, the participants disclose their preference/familiarity for introspection task, color/shape for object identification by utterance. This point is now clarified in 2.5.2.

Reference

  1. Kühn, S.; Fernyhough, C.; Alderson-Day, B.; Hurlburt, R.T. Inner experience in the scanner: can high fidelity apprehensions of inner experience be integrated with fMRI? Front. Psychol. 2014, 5, 1393. DOI:10.3389/fpsyg.2014.01393.

R2-2     Due to the task's complexity, a wide range of functional networks could have been involved in the task. The IFG's engagement, in particular, could be influenced more by verbal (motor) communications than by introspection effects. Was there a way to control the speech-related responses? As for the network of activation (of the introspection effects), the engagement of bilateral (anterior) temporal lobe and left angular gyrus regions seems very interesting, as shown in Fig. 3, but there are minimal inferences about those effects. These regions are known to be involved in semantic processes and could be the key selling point of the work, but they have received the least attention. Last but not least, the use of connectivity analysis confirms that the IFG is engaged in auditory and motor sensory regions and is less involved in semantic processes that are relevant to mental states. Perhaps other regions or a cluster of regions could be used as seeds for connectivity to confirm this.

Response:

As we compared introspection with object identification tasks, which included a similar amount of utterance (Figure 1 A and B), therefore the linguistic component should be effectively eliminated.

               We have discussed the bilateral temporal pole as the part of the subsystem of the default mode network, implicated in personal identity (last paragraph of 4.2), and the left angular gyrus, or left TPJ, as the part of the DMN, aka, mentalizing network (first paragraph of 4.1).

               The selection of the seed region for the PPI is based on the hypothesis that the region with the introspection-specific prediction error should show enhanced connectivity with regions with the lower-level prediction error (last paragraph 1. Introduction). As the introspection-specific mismatch-specific activity was found only in the left IFG, no other region was tested. No change was made in the text.

Minor comments,

R2-3     Please specify which brain atlas was used to report regions of interests.

Response:

We utilized Mai et al. (2015). This has been mentioned at the end of 2.7.2. No change was made in the text.

Reviewer 3 Report

The article provides an excellent theoretical framework. At some points in the contextualization, some of the references should be updated, since there have been significant advances. The irruption of Artificial Intelligence may represent a new turning point in terms of theoretical conceptualization. 

For example, in the section on predictive coding theory, it is suggested that the initial proposals be updated with others such as those of https://doi.org/10.1523/JNEUROSCI.3365-17.20185, which establish a necessary update. Also, studies such as those of https://doi.org/10.1073/pnas.171111411 or https://doi.org/10.1016/j.neuropsychologia.2019.107307 advocate a homogenization in the theoretical definition of the concept.

Table 1 should be adjusted to the width of the page to improve the comprehension and reading of the data presented.

Author Response

Reviewer 3

Comments and Suggestions for Authors

R3-1     The article provides an excellent theoretical framework. At some points in the contextualization, some of the references should be updated, since there have been significant advances. The irruption of Artificial Intelligence may represent a new turning point in terms of theoretical conceptualization.

For example, in the section on predictive coding theory, it is suggested that the initial proposals be updated with others such as those of https://doi.org/10.1523/JNEUROSCI.3365-17.20185, which establish a necessary update. Also, studies such as those of https://doi.org/10.1073/pnas.171111411 or https://doi.org/10.1016/j.neuropsychologia.2019.107307 advocate a homogenization in the theoretical definition of the concept.

Response:

Thank you very much for the suggestion of the updated references. We have updated the description in the Introduction (4th paragraph) with these inputs as follows:

The predictive coding theory (Friston, 2008; Mumford, 1992; Rao and Ballard, 1999) proposes that the comparison of the top-down signal of the mental model with the lower representation generates a prediction error, which is fed back up the hierarchy to update higher representations. This recursive exchange of signals suppresses the prediction error at every level to provide a hierarchical explanation for sensory inputs (Friston and Frith, 2015). The visual mismatch responses reflect the prediction errors obtained from a formal Bayesian model (Stefanics et al. 2018), supporting the theory. In language processing, predictive coding in the brain's response to language is domain-specific, which is sensitive to the hierarchical structure (Shain et al. 2020). The higher level of the hierarchical structure represents subjective experiences such as introspection (the capacity to attend to one’s thoughts and feelings) (Schilbach et al., 2012), which is represented by the mentalizing network (Andrews-Hanna, 2012). Thus, alignment represents the shared processing of the forward internal models at multiple levels between conversational partners, driven by prediction errors at each level. Through this hierarchical structure, the mismatch between what you and your partner uttered drives the conversation ultimately leading to understanding each other.

Shain, C.; Blank, I.A.; van Schijndel, M.; Schuler, W.; Fedorenko, E. fMRI reveals language-specific predictive coding during naturalistic sentence comprehension. Neuropsychologia 2020, 138, 107307.

Stefanics, G.; Heinzle, J.; Horváth, A.A.; Stephan, K.E. Visual mismatch and predictive coding: A computational single-trial ERP study. J. Neurosci. 2018, 38, 4020–4030.

R3-2    Table 1 should be adjusted to the width of the page to improve the comprehension and reading of the data presented.

Response:

We have adjusted the format accordingly.
